# Polypharmacy Management of Antipsychotics in Patients with Schizophrenia

**DOI:** 10.3390/medicina58111584

**Published:** 2022-11-03

**Authors:** Hiroyuki Kamei

**Affiliations:** Office of Clinical Pharmacy Practice and Health Care Management, Faculty of Pharmacy, Meijo University, Nagoya 468-8503, Japan; hkamei@meijo-u.ac.jp; Tel.: +81-528392643

**Keywords:** antipsychotics, polypharmacy, monotherapy, schizophrenia

## Abstract

Schizophrenia is a chronic psychiatric disease that is characterized by psychotic symptoms, including positive, negative, affective, and aggressive symptoms, as well as cognitive dysfunction, and is primarily treated using drug therapy, the continuation of which is essential to prevent recurrence/recrudescence. Various second-generation antipsychotics with pharmacological properties or adverse events that differ from those of conventional antipsychotics have recently been introduced, and pharmaceutical management is required for drug efficacy assessments and adverse event monitoring/management of these drugs. Antipsychotic monotherapy (APM) is the gold standard treatment for schizophrenia and is recommended in various guidelines. However, a subgroup of patients with schizophrenia do not or only partially respond to APM. Therefore, antipsychotic polypharmacy (APP), in which ≥2 antipsychotics are combined, has been routinely utilized to compensate for insufficient responses to APM in clinical practice. APP has recently been proposed as an evidence-based treatment option, but does not consider clinicians’ experience. However, the risk of APP-related adverse events is high. The application of APP needs to be carefully reviewed, whilst taking into consideration patient backgrounds. Furthermore, the risk of APP-related adverse events is higher in elderly patients than in the general population; therefore, caution is needed. This review discusses the merits of APP, matters that need to be considered, and a switch from APP to APM, and also focuses on the application of APP in clinical practice.

## 1. Introduction

Schizophrenia is a chronic psychiatric disease that frequently develops in patients aged 15 to 35 years [1]. Its primary symptoms are characterized by positive symptoms, such as hallucinations, delusions, and disordered speech and behavior; negative symptoms, including a lack of energy, apathy, isolation, self-neglect, and anhedonia; and a cognitive symptom domain that comprises impairments across different cognitive tasks, including executive functioning, attention and information processing, affective symptoms, and aggressive symptoms [2]. The long-term goals of treatment for this disease are the prevention of recurrence/recrudescence and rehabilitation. The continuation of drug therapy with antipsychotics is essential for the attenuation of psychotic symptoms in patients with schizophrenia and the prevention of recurrence/recrudescence. Many antipsychotics have been developed and are used to treat patients with schizophrenia. However, the outcome of treatment for schizophrenia in clinical practice is still not satisfactory [3,4,5]. Antipsychotic monotherapy (APM) is the gold standard treatment for schizophrenia and is recommended in various guidelines [6,7,8,9,10]. However, between 10 and 60% of patients with schizophrenia do not or only partially respond to APM [11,12,13]. Although antipsychotic polypharmacy (APP) is not recommended in major treatment guidelines, it has been routinely utilized to compensate for insufficient responses to APM in clinical practice. APP is defined as the use of ≥2 antipsychotics to treat schizophrenia [14,15].

APP is proposed as a treatment option after APM-related failures [16,17,18]. Furthermore, antipsychotic dose/drug reductions have recently been proposed to manage APP-related adverse events [19,20,21]. The adverse events associated with APP need be considered for elderly patients [22]. Although limited information is currently available on the therapeutic efficacy of APP in elderly patients, it was previously shown to be similar to that in the general population [23,24].

In the light of the discrepancies outlined above, this narrative review highlights the gap that currently exists between treatment guidelines and practice for the use of APM or APP therapy in patients with schizophrenia. This review is based on the literature collected from sequential searches in PubMed from January 2012 to August 2022, reference lists in existing reviews and papers on the main topic of antipsychotic polypharmacy or antipsychotic combination treatment. The search was based on the following words or Medical Subjects Heading terms: “schizophrenia” and “antipsychotics” or “monotherapy” or “polypharmacy” or “combination treatment” or “individual-drug name”, and “randomized controlled trials” or “meta-analysis” Therefore, the review was not performed using standard search criteria or methods for a systematic review. During the process, screening and selection of publications were carried out regarding which contents and data were reviewed and presented. The search was limited to original English-language articles published in peer-reviewed journals. An effort was made to preferentially include studies from secondary mental health services in order to ensure comparability between included study populations. The review specifically focuses on secondary literature that investigates the outcome of antipsychotic monotherapy and combination treatment. Thus, this review provides several practical points that need to be considered for the utilization of APP in routine medical care.

## 2. Proper Use of Antipsychotics

Antipsychotics, such as quetiapine, perospirone, olanzapine, aripiprazole, brexpiprazole, blonanserin, clozapine, paliperidone, and asenapine, have become commercially available, starting with the second-generation antipsychotic (SGA) risperidone. Since recurrence and adverse event-related dropout rates are low with these antipsychotics, they are recommended as first-line drugs in several guidelines [2,20,25]. The recommended dosages and chlorpromazine equivalent doses (CPZeq) of selected SGAs are shown in Table 1 [26,27,28]. The profiles for each receptor (Table 2) [29,30,31] and adverse events (Table 3) [32] vary among SGAs; therefore, it is important to select adequate antipsychotics in accordance with individual patient backgrounds and establish dosage and administration regimes, while monitoring improvements in psychotic symptoms and the development of adverse events. Meta-analyses of the utility of antipsychotics for the treatment of schizophrenia showed that the effect sizes of SGAs other than clozapine were similar to those of first-generation antipsychotics (FGAs) [33,34]. In comparison with a placebo, standardized mean differences (SMDs) with 95% credible intervals (CrI) for overall changes in symptoms were as follows: clozapine −0.89, −1.08 to −0.71; olanzapine −0.56, −0.62 to −0.50; risperidone −0.55, −0.62 to −0.48; paliperidone −0.49, −0.59 to −0.38; haloperidol −0.47, −0.53 to −0.41; quetiapine −0.42, −0.50 to −0.33; aripiprazole −0.41, −0.50 to −0.31; asenapine −0.39, −0.52 to −0.26; and brexpiprazole −0.24, −0.53 to −0.05. In comparison with a placebo, SMDs for significant reductions in positive symptoms ranged from −0.61 (95% CrI −0.68 to −0.54) for risperidone to −0.17 (−0.31 to −0.04) for brexpiprazole [34]. Olanzapine, paliperidone, and haloperidol were significantly more effective than many other drugs [34]. In comparison with a placebo, SMDs for significant reductions in negative symptoms ranged from −0.62 (95% CrI −0.84 to −0.39) for clozapine to −0.25 (−0.36 to −0.14) for brexpiprazole. Clozapine, olanzapine, and, to a lesser extent, risperidone reduced significantly negative symptoms more than many other drugs [34].

On the other hand, concerning adverse reactions, the characteristics of each drug are shown in Table 3 [32,35]. The characteristics of each drug shown in Table 3 are based on relative comparisons rather than their absolute properties. Therefore, it is important to note that any drug may have an adverse event in an individual patient. Sedation may be a therapeutic target in the acute treatment of patients that present with agitation or severe behavioral symptoms. Sedation is linked to the blockade of histaminergic receptors and is the highest for clozapine, followed by quetiapine, olanzapine, haloperidol, asenapine, and risperidone. Antipsychotic agents are generally associated with weight gain; thus, care is needed when they are administered to patients with diabetes mellitus. Although the risk of weight gain is low with the majority of high-potency FGAs, low-potency FGAs and most SGAs markedly increase the risk of weight gain and, ultimately, obesity in patients with schizophrenia [36]. However, weight gain potential markedly differs among SGAs, and some FGAs may result in more weight gain than specific SGAs [36]. Clozapine and olanzapine markedly increase body gain and the risk of developing diabetes mellitus. Extrapyramidal symptoms (EPS), including bradykinesia, muscle rigidity, tremors, dystonia, akathisia, and tardive dyskinesia, are linked to the ratio of D_2_ receptor to 5HT_2A_ receptor binding [37]. The highest incidence of EPS in patients with schizophrenia occurs with haloperidol, with moderate EPS being observed with risperidone, paliperidone, and milder EPS with asenapine, aripiprazole, and brexpiprazole. Akathisia is defined as a compelling need for constant motion that is associated with marching up and down and crossing and uncrossing the legs when sitting [38]. It is an adverse event caused by both FGAs and SGAs; however, a meta-analysis reported that FGAs were more likely to cause clinically relevant akathisia [39]. Comparative meta-analyses showed that aripiprazole induced akathisia more frequently than olanzapine in patients with schizophrenia and clozapine and risperidone had moderate effects [37,40]. It is important to note that some overlap between akathisia and pseudo-akathisia labeling may have influenced these findings. A large population-based study identified dry mouth and constipation as anticholinergic adverse events that were the most frequently associated with the administration of clozapine, olanzapine, quetiapine, and low-potency FGAs and increased the risk of dental caries [41]. Sialorrhea is a frequent and paradoxical adverse event caused by clozapine [42]. Hypersalivation markedly impairs quality of life and may interfere with social functioning. The degree of hyperprolactinemia depends on D_2_ receptor occupancy, as well as the antagonist properties of antipsychotics [43]. Antipsychotics with a higher D_2_ affinity and antagonist properties, namely haloperidol, risperidone, and paliperidone, have been shown to markedly increase serum prolactin levels [43]. Mild hyperprolactinemia developed in patients treated with olanzapine and asenapine, but not in those administered quetiapine or clozapine. In contrast, partial D_2_ agonists, such as aripiprazole and paliperidone, were found to lower prolactin levels, even below the drug-free baseline, while adjunctive aripiprazole decreased hyperprolactinemia associated with other antipsychotics [44].

Prior to the introduction of long-acting injections (LAI) of antipsychotics, it is important to administer an oral preparation of the same drug in order to confirm tolerance (concerning paliperidone palmitate, risperidone is available). However, since titer conversions from an oral preparation do not apply to all patients, care is needed to prevent overdoses when switching to LAI or unexpected adverse events [45]. The inhaled antipsychotic loxapine represents a novel option in the acute treatment of agitation in patients with schizophrenia that combines the rapid onset of effects with a non-invasive route of administration [46]. Although it is easy to self-administer, inhaled loxapine requires a degree of cooperation from the recipient, and thus is not a substitute for an injection during psychiatric emergencies, when a patient is actively refusing medication [46]. The efficacy and safety of inhaled loxapine in elderly patients and in outpatient care settings have not yet been established. A blonanserin transdermal patch, which also offers potential benefits, including improved adherence, was previously shown to attenuate the symptoms of acute schizophrenia with acceptable tolerability, and the most common adverse events were erythema and pruritus at the site of application [47]. Extended-release paliperidone is a new atypical antipsychotic that is chemically related to risperidone. It has been formulated in an osmotic controlled-release oral delivery system that minimizes peak–trough fluctuations and, by obviating dose titrations, it allows for once-daily dosing with a therapeutically active dose from the first day [48]. However, long-term treatment with extended-release paliperidone and olanzapine resulted in significant increases in weight gain and waist circumference [49,50]. These findings reinforce the necessity of regularly monitoring metabolic parameters in patients with schizophrenia treated with atypical antipsychotics, including extended-release paliperidone.

## 3. Polypharmacy

### 3.1. Factors for Polypharmacy

In a recent systematic review [34], regional differences were observed in the rate of APP; those in North America, Oceania, Europe, and Asia were 16, 16.4, 23, and 32%, respectively. Furthermore, the average daily dose of antipsychotics in 15 Asian countries was 424 ± 376 mg (CPZeq) [32]. According to an extensive meta-analysis, the rate of APP with 2 antipsychotics ranged between 17.8 and 44.1%, while that with ≥3 antipsychotics was between 0.2 and 24.4% [34,35,36].

Figure 1 shows clinical steps in APP. APM, particularly SGAs, is recommended in various guidelines [6,7,8,9,10]. Antipsychotics are the basic drug of choice to treat the psychotic symptoms of schizophrenia. However, many patients with schizophrenia only partially respond (i.e., the persistence of symptoms such as delusions and hallucinations) to a standard dose of an initially prescribed antipsychotic drug. In these cases, clinicians may increase the antipsychotic dose beyond regular thresholds or switch to a different antipsychotic drug in order to enhance antipsychotic efficacy.

Regarding switching, no marked differences were observed between increasing the dose of an antipsychotic and switching to a different antipsychotic. Limited evidence is currently available and is of very low quality. The guidelines originally recommended waiting for four to eight weeks before switching to another drug, arguing that the full efficacy of a given drug is reached after a longer period of treatment [51]. However, recent findings suggest that non-responders may be identified as early as two weeks after the initiation of treatment [52]. A previous study estimated that between one-fifth and one-third of patients with schizophrenia did not respond adequately to standard antipsychotic treatment [53]. Although aripiprazole is suggested to be highly beneficial, particularly in patients in whom adverse events to other drugs made the continuation of drug therapy difficult, it does not exert anticholinergic effects; therefore, anxiety or impatience suggestive of choline rebound may develop when switching from other drugs. Similarly, a method to switch from other drugs to asenapine, which does not exert anticholinergic effects, also needs to be reviewed.

Clinicians combine antipsychotics with an antidepressant for negative symptoms, benzodiazepines for comorbid anxiety or distress, or a mood stabilizer for incapacitating mood instability [54]. In addition, some clinicians combine several antipsychotic drugs to achieve superior therapeutic effects and attenuate adverse events. However, most of the clinician-argued reasons for antipsychotic combination treatment lack a clear rationale and the documentation of therapeutic benefits [54]. Although no concrete criteria for APP have been established, it was recently defined as the use of ≥2 antipsychotics primarily for the treatment of schizophrenia [8]. Since the efficacy of APM is insufficient in patients with schizophrenia, clinicians need to primarily attempt to ameliorate positive and/or negative symptoms, particularly positive symptoms, by utilizing APP. Furthermore, APP is used to treat specific concomitant symptoms, such as anxiety, cognitive dysfunction, impulsive/aggressive behavior, and sleep disturbance. In addition, the reasons why APP is selected include the prevention of recurrence/recrudescence, the avoidance of high dosages by APM, duplication when switching to monotherapy, shortening of the admission period, the inhibition of re-admission, the promotion of treatment responses, and the prevention of adverse events based on the different profiles of affinity for various receptors [55]. In addition, the appearance of antipsychotics with various pharmacological profiles (Table 2) has increased the number of drug options, as well as the number of APP variations. Clozapine is the only drug indicated for refractory schizophrenia, but may induce the serious adverse event agranulocytosis; therefore, the rate at which this drug is used is extremely low [10,56,57]. In the future, the widespread use of clozapine as a type of general drug therapy needs to be promoted [58,59].

### 3.2. Merits of Polypharmacy

The merits of polypharmacy in the treatment of schizophrenia have recently been emphasized, despite the concerns associated with APP. A meta-analysis [60] of data from 16 studies that compared APP with APM in patients with schizophrenia showed that APP reduced all psychotic symptoms, with a marked difference in the effect size (SMD −0.53, 95% confidence interval (CI) −0.87–−0.19), indicating the superiority of APP to APM. However, 14 high-quality, double-blind studies did not show the superiority of APP to APM for attenuating psychotic symptoms^38)^. In addition, the superiority of APP was not observed when study-defined multiple response rate criteria were used; however, it was noted in a previous meta-analysis study with a smaller sample [61]. Briefly, combination therapy with aripiprazole reduced negative symptoms (SMD −0.41, 95% CI −0.79–−0.03), but no superiority was recorded with respect to discontinuation, global clinical impressions, or positive/general/depressive symptoms (eight studies) [61]. In this study, no significant differences were observed in the appearance of adverse events between APP and APM. Combination therapy with D2 receptor antagonists attenuated insomnia, while therapy with aripiprazole decreased prolactin levels and body weight [61].

In a nationwide study in Hungary [62], APP (by adding the second antipsychotic after APM for ≥60 days) was compared with APM (by switching to a new antipsychotic after APM for ≥60 days), and the rate of patients who were admitted to the psychiatric hospital was significantly higher in the APM group than in the APP group (hazard ratio (HR) 1.69). In addition, a nationwide cohort study [63] that involved 62,250 patients with schizophrenia analyzed 29 different APM and APP treatment types for 10 years, and indicated that the risk of psychiatric readmission was lower in patients treated by APP with clozapine and aripiprazole than in those treated by APM with clozapine, which was the lowest in the cohort. Furthermore, a 14% (HR 0.86) difference was noted between the two groups. A randomized controlled trial (RCT) (n = 127) for 6 months [64] evaluated the clinical merits and risks of the continuation of APP and switching to APM at an outpatient clinic, and showed that the interval until all-cause discontinuation, as a primary endpoint, was longer in the APP continuation group than in the APM-switched group; the discontinuation of treatment was more frequent in the latter than in the former.

No marked differences were detected in the efficacy of APP among older adults and the general population. However, this finding was based on small-scale studies that involved a small number of patients, not RCTs or large-scale studies [23,24,65].

### 3.3. Matters to Be Considered with Polypharmacy

APP is clinically associated with some disadvantages, including increases in the frequency or severity of adverse events, drug interactions, and treatment complexity-related medication errors and a reduction in adherence to medication [66,67,68,69]. Moreover, readmission rates were higher with APP than with APM, which is contradictory to the findings of the studies described above. A recent study in the UK showed that the risk of readmission was significantly higher in patients discharged under APP than in those discharged under APM (HR 1.4, 95% CI 1.2–1.7) [70]. This risk was markedly higher in patients discharged under APP with clozapine (HR 1.8, 95% CI 1.2–2.6) [70]. A recent review/meta-analysis reported that APP with D2 antagonists was associated with a higher incidence of hyperprolactinemia, EPS, sexual dysfunction, hypersalivation, sedation/somnolence, cognitive dysfunction, and diabetes mellitus, as well as greater weight gain and the more frequent use of anticholinergic drugs than APM [60]. It is important to consider the appearance of these APP-related adverse events and the use of anticholinergic drugs in elderly patients. A survey on 2500 subjects aged ≥ 65 years in the UK showed that the use of anticholinergic drugs increased the risk of cognitive dysfunction and death (odds ratio 1.56, 95% CI 1.36–1.79) [71].

APP, namely high-dose antipsychotic therapy, markedly deteriorated cognitive function regardless of the type of antipsychotic, SGA or FGA, that was added to the first antipsychotic [39,72]. Furthermore, the average daily dose of antipsychotics was associated with the onset of cognitive dysfunction in patients with schizophrenia [73]. Briefly, APP was strongly associated with a higher daily dose (12.1 mg/d of a risperidone equivalent dose; RISeq) and a greater reduction in cognitive function measured using the z score of the Brief Assessment of Cognition in Schizophrenia, a scale used to assess cognitive function, in comparison to APM (4.2 mg/d of RISeq) [73]. Moreover, an improvement in cognitive function measured using the Wisconsin card sorting test (the total number of correct answers increased by 19.9%, while that of errors decreased by 34.9%) was achieved when the doses of antipsychotics were reduced in schizophrenia patients treated with high-dose APP [74]. These findings indicate that the total dosage of antipsychotics has important implications for the amelioration of cognitive dysfunction, which may be related to rehabilitation in schizophrenia patients [75]. On the other hand, a previous study reported that cognitive hypofunction was less frequent in patients treated with high-dose antipsychotic therapy (higher than 1000 mg/day of CPZeq) or APP than in those treated with low-dose antipsychotic therapy or APM, suggesting that high-dose therapy or APP does not always cause cognitive hypofunction [76]. However, findings on the causal effects of APP on cognitive function are still inconsistent.

According to a previous study [77], recurrence was detected within 1 year in approximately ≥50% of schizophrenia patients who received acute-phase treatment and within 5 years in approximately 80%. In 85% of patients with recurrence, it was observed twice or more. A reduction in adherence to medication has been suggested as a contributing factor. Poor adherence to medication is a major barrier to achieving an optimal clinical outcome in patients with schizophrenia. Polypharmacy is associated with poor adherence to medication for many reasons; APP may affect the continuation of treatment [78]. A recent study reported that the medication non-adherence rate in patients with schizophrenia was 41.0%, with the risk of non-adherence being approximately 2-fold higher in patients treated with APP than in those receiving APM [79]. It currently remains unclear whether the number of antipsychotics is directly associated with a reduction in adherence to medication; however, APP was strongly associated with an increase in the risk of various adverse events that led to a reduction in the persistency of medication adherence [80]. Therefore, APP may, at least indirectly, increase the risk of medication non-adherence and non-persistency, thereby influencing the clinical course and outcomes of treatment.

## 4. Strategies for Polypharmacy

### 4.1. Switching to APM

Current symptoms and the clinical state of patients need to be clarified prior to the initiation of APP. Follow-ups after the start of APP are essential. If symptoms are not ameliorated or if it is impossible to continue treatment due to the appearance of adverse events, APP needs to be switched to APM or other combinations must be considered [81]. If symptoms are attenuated and a stable condition is achieved, the switch to APM needs to be carefully monitored [64,82].

A study on dose/drug reduction from APP to APM reported an increased dropout rate after a switch to APM in schizophrenia patients that took two antipsychotics in the United States [64]. Briefly, the treatment continuation rate was compared for 6 months in 127 patients who had taken two antipsychotics (358 mg/d, CPZeq) and were randomly assigned to one of the following two groups: a group in which two-drug therapy was switched to monotherapy within 1 month (switch group) and a group in which two-drug therapy was not switched (stay group). The dropout rate in the switch group was 31%, which was significantly higher than that in the stay group (14%) [64]. In an open-label study in Japan, APP was switched to APM within 6 months in 44 schizophrenia patients that took 29 antipsychotics (1109 mg/d, CPZeq) on average, and psychotic symptoms deteriorated in 22.7% of the patients [83]. These findings suggested that rapid dose reductions during APP contributed to symptom deterioration, and an examination using the safety correction of high-dose antipsychotic polypharmacy (SCAP) method was recently conducted [84,85]. In other words, the concrete rates of dose-reduction were proposed [86,87]: among antipsychotics, regarding drugs with an amount of <10 mg equivalent to 100 mg of CPZ as high-titer drugs, the dose is decreased by 50 mg on CPZ conversion in at least 1 week; and, regarding drugs with an amount of ≥10 mg equivalent to 100 mg of CPZ as low-titer drugs, the dose is decreased by 25 mg on CPZ conversion in 1 week. To investigate the safety of the SCAP method and the absence of symptom deterioration, an open-label, multicenter, cooperative RCT that involved 163 schizophrenia patients that took ≥2 antipsychotics (500–1500 mg/d, CPZeq) was planned in Japan, and the findings obtained were compared between the antipsychotic reduction and control (non-antipsychotics reduction) groups. A 0.5–drug reduction, on average, with a 23% dose reduction as a CPZ-converted titer was achieved in the drug-reduction group during a period of 6 months. No significant differences were observed in psychotic symptoms or adverse events between the two groups and no serious adverse events occurred [67,68]. A recent study reported that among patients in whom drug/dose reduction was performed using the SCAP method in clinical practice, switching to monotherapy was safely and successfully achieved without the deterioration of psychotic symptoms within 6 months in 5 schizophrenia patients that took 2.4 antipsychotics (700 mg/d, CPZeq) on average, and the degree of patient satisfaction was high [81]. Even among patients with adverse events caused by APP and that strongly requested antipsychotic drug/dose reductions, many patients felt anxious about these drug reductions. Therefore, when performing antipsychotic drug/dose reductions using the SCAP method, it is important to sufficiently and repeatedly explain symptom management methods and a potential return to the original dose at any time to patients/their family members with anxiety, regarding the drug reduction-related deterioration of psychotic symptoms or the appearance of withdrawal symptoms [81]. Another study identified an age of ≥40 years, disease duration of ≥10 years, and CPZeq of ≥200 mg/day after dose reductions as important patient factors for successfully achieving dose/drug reductions following APP [88]. In the future, the pharmacological characteristics and pharmacokinetics/-dynamics (PK/PD) of individual antipsychotics need to be considered, in addition to patient backgrounds, in order to perform dose/drug reductions from APP to APM more precisely in clinical practice. Concerning APM, as the medical costs and risks of adverse events are both low, physicians must make efforts to promote APM in most patients [86,87]. Moreover, patients only need to remember to take a single antipsychotic; thus, adherence to APM may be more favorable than that to APP [65].

### 4.2. Practical Use of Dosage Forms

As a strategy used to improve adherence to medication in patients with schizophrenia, the practical use of dosage forms has been emphasized [89]. Various dosage forms of antipsychotics, such as tablets, powders, orally disintegrating tablets, liquids for internal use, sublingual tablets, and patches, have recently become commercially available, thereby facilitating the selection of patient preference-matched dosage forms [90]. LAI (risperidone, paliperidone palmitate, and aripiprazole) is a type of controlled-release preparation designed to achieve a stable blood concentration with administration at 2- to 12-week intervals. Administration by medical staff in outpatient visits facilitates the continuation of antipsychotic treatment, making it possible to avoid discontinuation-related recurrence/recrudescence. The findings of a large-scale clinical trial in Spain showed that the two-year treatment continuation rate for oral drugs was 63%, whereas that for LAI was 82%, which was high [91]. Furthermore, poor LAI adherence is a clear indicator of irregular or discontinued hospital visits, facilitating adherence assessments in patients; therefore, early interventions before recurrence are possible, which is also useful to note. A cohort study on 29,823 patients with schizophrenia in Sweden indicated that the risk of readmission after treatment with LAI (paliperidone, olanzapine, and risperidone) was the lowest, and was similar to that with clozapine [92]. From a pharmacokinetic viewpoint, LAI is not influenced by intestinal absorption and there is no first-pass effect in the liver after intramuscular injection; therefore, there are no blood concentration changes related to individual differences in metabolic enzyme activity, in contrast to oral drugs [93]. On the other hand, there are LAI-specific demerits. Injection-site reactions, such as pain, swelling, pruritus, and induration, which may be induced by injections, are adverse events that are not observed after the administration of oral drugs. Furthermore, when LAI is administered, the drug cannot be promptly excreted; therefore, even when administration is discontinued due to adverse events, the drug may remain in the body for a long period, negatively impacting/delaying symptoms.

In a recent small-scale study, the administration of aripiprazole once monthly (AOM) was introduced for APP patients in clinical practice. The mean number of antipsychotics before the introduction of AOM was 2.4 drugs, but markedly decreased to 0.7 12 months later [94]. In addition, psychotic symptoms (PANSS: −13.6%, CGI-S: −8.8%) were attenuated and adverse events, such as EPS, also decreased [94]. These findings suggest that LAI is advantageous for preventing recurrent schizophrenia and correcting high-dose APP, as demonstrated for clozapine. In the future, the widespread use of LAI may serve as a strategy to overcome APP. Since LAI is an invasive dosage form, a sufficient explanation and the proper confirmation of intention from patients themselves before its introduction are required at a higher level [95]. To achieve this, the selection of treatment by “shared decision making”, in which treatment is selected from two directions, medical staff and the patient, is necessary [96]. It is important to provide information on the merits/demerits of LAI to patients, whilst taking into account their lifestyle or values. The efficacy and safety of LAI APM appeared to be similar to those of the combination of LAI and other oral antipsychotics in patients with schizophrenia. Therefore, the combination of LAI and oral antipsychotics, which is commonly used in clinical practice, may not be necessary [97].

## 5. Conclusions

APM is recommended as the gold standard of treatment for schizophrenia, regardless of its stage in clinical practice. However, many patients do not or only partially respond to APM at a sufficient dose. Recent studies described the merits and limitations of APP, which is recommended as an option for non-responders to APM, as well as strategies to overcome these limitations. In any case, it is necessary to understand the pharmacological characteristics of various antipsychotics/expected adverse events and carefully review the use of APP, whilst taking into consideration drug information based on experience regarding their use in individual patients. Prior to the introduction of APP for elderly patients, attention must be paid to various adverse events related to antipsychotics and to concomitant physical diseases, aging-related physiological hypofunction, and combined drug interactions. Furthermore, even if APP is initiated, it cannot continue for an unspecified length of time. Physicians must promote APM, while precisely monitoring the balance between treatment effects and adverse events. Strategies to avoid APP include antipsychotic dose/drug reduction using the SCAP method and the practical use of dosage forms, such as LAI. The final goal of schizophrenia treatment is to achieve rehabilitation [98]. The continuation of drug therapy is the most important strategy for preventing recurrence, readmission, suicide attempts, or impulsive behaviors and maintaining improvements. Further evidence of the beneficial use of APP for patients with schizophrenia, including elderly patients, is needed in the future.

## Figures and Tables

**Figure 1 medicina-58-01584-f001:**
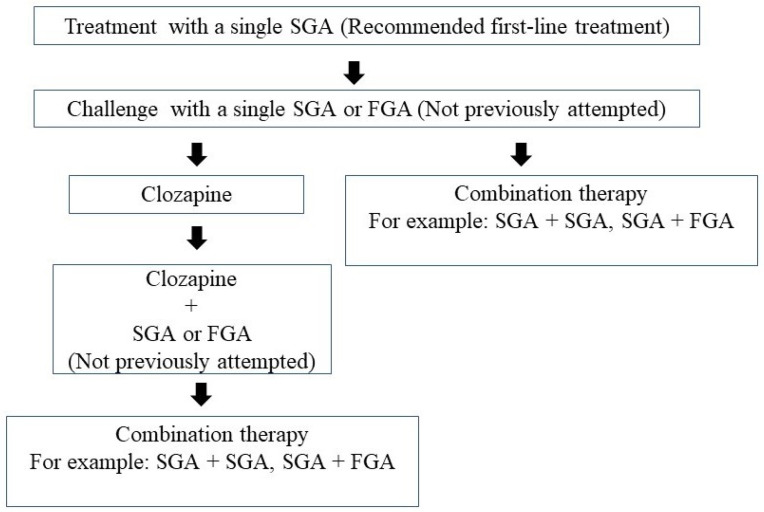
Clinical steps in antipsychotic polypharmacy (APP). FGA: first-generation antipsychotic, SGA: second-generation antipsychotic.

**Table 1 medicina-58-01584-t001:** Dosages and chlorpromazine equivalent doses (CPZeq) of selected second-generation antipsychotics. [26].

	Dose Range (mg/Day)	CPZeq (mg)
Risperidone	2–16	1
Paliperidone	1.5–15	1.5
Aripiprazole	2–30	4
Brexpiprazole	1–4 a	-
Asenapine	0.4–20	2.5
Quetiapine	75–800	66
Olanzapine	1–20 b	2.5
Clozapine	100–600	50
Haloperidol (First-generation antipsychotic)	4–20	2

CPZeq indicates an equivalent to 100 mg of CPZ, -; Not yet defined, a: [27]; b: [28].

**Table 2 medicina-58-01584-t002:** Comparison of receptor affinities of selected second-generation antipsychotics [29,30,31].

Drug	Receptor Affinity							
	Dopamine	Serotonin			Adrenaline		Histamine	Muscarine
	D_2L_	5-HT_1A_	5-HT_2A_	5-HT_2C_	α_1A_	α_2B_	H_1_	M_1_
Risperidone	6.17	178	0.204	6.76	5.13	9.55	81.3	26,900
Paliperidone	6.60	1030	1.20	19.0	2.50	4.00	19.00	>10.000
Aripiprazole	1.15	2.69	9.55	28.2	324	191	20.4	3890
Brexpiprazole	0.30	0.12	0.47	34.0	3.80	35.0	19.0	>1.000
Asenapine	1.26	2.51	0.0708	0.0347	1.17	0.324	1.00	8130
Quetiapine	417	166	155	1050	64.6	83.2	11.0	282
Olanzapine	21.4	1510	1.32	3.89	22.4	331	3.39	12.0
Clozapine	135	87.1	4.07	2.75	12.6	28.2	1.74	5.13
Haloperidol (First-generation antipsychotic)	1.45	513	52.5	1620	25.1	562	2090	5620

Values represent Ki values (nM). Risperidone and paliperidone primarily exert antagonistic effects on serotonin/dopamine receptors. Aripiprazole and brexpiprazole primarily act as dopamine receptor partial agonists. Asenapine shows high affinity for serotonin/adrenaline receptors, in addition to dopamine receptors. Quetiapine, olanzapine, and clozapine act on several receptors, including serotonin/dopamine receptors.

**Table 3 medicina-58-01584-t003:** Comparison of adverse events to selected second-generation antipsychotics [32,35].

Drug	Sedation	Weight Gain	Diabetes Mellitus	Extrapyramidal Symptoms	Anticholinergic Effects	Increase in Prolactin Levels
Risperidone	+	++	++	+	+	+++
Paliperidone	+/−	+/−	−	+	+	+++
Aripiprazole	−	+/−	−	+	−	−
Brexpiprazole	−	+/−	−	+	−	−
Asenapine	+	+/−	−	+	−	+/−
Quetiapine	++	++	++	−	−	−
Olanzapine	++	+++	+++	+/−	+	+
Clozapine	+++	+++	+++	−	+++	−
Haloperidol (First-generation antipsychotic)	+	+	+	+++	−	+++

+++: High frequency/severe, ++: medium frequency/moderate; +: low frequency/mild.

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
