# Peer review of "Polypharmacy Management of Antipsychotics in Patients with Schizophrenia"

_medicina, 2022, doi:10.3390/medicina58111584_

Round 1

Reviewer 1 Report

Thank You for this interesting review about APM and APP.  This is a significant topic that is not often highlighted in medical training or clinical approach from an evidence based approach.  Just recommending some small edits and a couple of additional citations.  It is clear that you worked hard to write cohesively and a piece that could benefit readers. I have uploaded a PDF with comments (click the comments function on the right) for review. 

Author Response

Responses to Reviewer 1 comments:

I wish to express our appreciation to the Reviewer for his or her insightful comments, which have helped me to significantly improve the quality of our manuscript.

Comment 1: These generalized characterizations are helpful but not absolute, recommend editing the language to reflect that any of these medications may be associated with any of the noted ADRs for the antipsychotics class of drugs in the individual patient.  But, these are general characteristics for these medications. (What's not adding up for me for example is that aripiperazole is known for less EPS due to lower dopamine % time bound as an effect of it's 5HT affinity)

Response: In accordance with the Reviewer’s comment, we added the following sentence to the 2. Proper use of antipsychotics section: “The characteristics of each drug presented in Table 3 are based on relative comparison rather than absolute properties of the drug. Therefore, it should be mentioned that any drug has at least all adverse reactions in the individual patient.” (Page 2 line 39 – line 41)

Comment 2: This sentence would read better if it is designated as a class effect (English writing synthesis) "In general, antipsychotic agents are associated with weight gain, and are (recommend not using contraindicated- rather should be used with caution) for patients with diabetes mellitus.  

Response: In accordance with the Reviewer’s comment, I changed “However, weight gain frequently appears, and these drugs are contraindicated for patients with diabetes mellitus” to “In general, antipsychotic agents are associated with weight gain, and are caution for patients with diabetes mellitus.” (Page 2 line 44 – line 46)

Comment 3: Recommend editing "as an issue" from an English grammar perspective, this sentence would read better if ending after "should be reviewed" no need to add "as an issue"

Response: In accordance with the Reviewer’s comment, I removed the "as an issue" (Page 5 line 22 – line23)

Comment 4:  Include citation for resource where the table was adapted from.

Response: I added the reference number [29-31] in Table 2. (Page 4)

Comment 5:  Recommend labeling "Receptor Affinity"

Response: I changed “Receptor” to "Receptor Affinity" in Table 2. (Page 4)

Comment 6: Include citation for resource where the table was adapted from.

Response: I added the reference number [35,36] in Table 3. (Page 3)

Comment 7: Recommend adding a citation here

Response: I added a citation as follows; [53] Reynolds. G.P.; High dose antipsychotic polypharmacy and dopamine partial agonists - time to rethink guidelines? J Psychopharmacol. 2021, 35, 1030-1036. (Page 3 line 13 - line 19)

Comment 8: APP

Response: I changed “this” to "APP". (Page 6 line 19)

Comment 9:  Recommend not using the term elderly (not recommended by American Geriatrics Society), rather use "older adults"

Response:  I changed “elderly” to " older adults ". (Page 6 line 44)

Comment 10: Is there a citation that endorses this recommendation?

Response: I added a citation as follows; [80] Kamei, H.; Yamada, H.; Hatano, M.; Hanya, M.; Yamada, S.; Iwata, N.; Effectiveness in Switching from Antipsychotic Polypharmacy to Monotherapy in Patients with Schizophrenia: A Case Series. Clin. Psychopharmacol. Neurosci. 2020, 18, 159-163. (Page 8line 5)

Reviewer 2 Report

Although the author tried to elaborate on the clinical issues of antipsychotic polypharmacy, this paper needs a major revision to achieve this goal. 

Specific comments:

⁃ The Title: Consider revising to reflect the manuscript's content, emphasizing the antipsychotic polypharmacy, because other psychotropic drugs can also be given in this indication. 

⁃ The Abstract: The Authors should incorporate a description of 5 domains in the first sentence. Should also be mentioned the therapy resistance (nonresponse), how the antipsychotics could be combined, and the benefits of combining antipsychotics.

⁃ Keywords: Consider excluding “elderly patients” because it is not one of the major points and is only shortly described, maybe to include therapy resistance or nonresponse.

⁃ The Introduction: Explain 5 domains (Stahl: positive, negative, cognitive, affective, and aggressive symptoms). Announce that it will be in the scope why, when, and how antipsychotics will be combined.

⁃ Proper use of antipsychotics: Ad one more table at the beginning with dose ranges and equivalent doses of all mentioned antipsychotics. In all tables should be added frequently used typical antipsychotics. This subsection is better to be divided into different paragraphs. Firstly, the individual profile of action of antipsychotics with their clinical properties needs to be more thoroughly commented on in the text, not only in the table legend. Secondly, present the possible adverse effects. Thirdly, about switching strategies. 

⁃ Polypharmacy: Explain treatment resistance (nonresponse). Other options for augmentation with psychotropic drugs before APP. When is APP justified and in which combinations (FGA/SGA)? 

⁃ Switching to APM: Consider excluding or transferring into the Discussion the following part: “Concerning APM, generally, the costs… than that to APP (63).”

⁃ Practical use of dosage forms: It is important to comment on the combination of LAI and other AP. Transfer the part about formulations in the section Proper use of antipsychotics and add to all of this about prolonged-release tablets and inhaling devices.

⁃ Conclusion: Should be rewritten following all changes made. It should reflect the significant points of the manuscript. 

⁃ There are a lot of newer references in the field. 

⁃ Add a figure reflecting clinical steps in APP. 

Author Response

Responses to Reviewer 2 comments:

I wish to express our appreciation to the Reviewer for his or her insightful comments, which have helped me to significantly improve the quality of our manuscript.

Comment 1: The Title: Consider revising to reflect the manuscript's content, emphasizing the antipsychotic polypharmacy, because other psychotropic drugs can also be given in this indication. 

Response: In accordance with the Reviewer’s comment, I changed the title “Polypharmacy management in patients with schizophrenia” to “Polypharmacy management of antipsychotics in patients with schizophrenia” (Page 1)

Comment 2: The Abstract: The Authors should incorporate a description of 5 domains in the first sentence. Should also be mentioned the therapy resistance (nonresponse), how the antipsychotics could be combined, and the benefits of combining antipsychotics.

Response: In accordance with the Reviewer’s comment, I added “affective symptoms, and aggressive symptoms.” in the first sentence. However, regarding “therapy resistance (nonresponse), how the antipsychotics could be combined, and the benefits of combining antipsychotics”, the last sentence includes most of those implications. (Page 1 Abstract: line 2-line 3)

Comment 3: Keywords: Consider excluding “elderly patients” because it is not one of the major points and is only shortly described, maybe to include therapy resistance or nonresponse.

Response: In accordance with the Reviewer’s comment, I removed the "elderly patients”. (Page 1; Keywords)

Comment 4: The Introduction: Explain 5 domains (Stahl: positive, negative, cognitive, affective, and aggressive symptoms). Announce that it will be in the scope why, when, and how antipsychotics will be combined.

Response: In accordance with the Reviewer’s comment, I added “affective symptoms, and aggressive symptoms.” (Page 1 line 7 - line 8) And I added Figure 1regarding the algorithm of combination therapy of antipsychotics in the section of 3. Polypharmacy. (Page 1; 1. Introduction: line 6-line 7)

And also I added the Figure1“Clinical steps in APP” (Page 5) and the comment regarding why, when, and how antipsychotics will be combined as follws;

“Figure 1 shows clinical steps in APP. First, antipsychotic monotherapy (APM), especially SGA is recommended in various guidelines [6-10]. Antipsychotics are the basic drug of choice for treating the psychotic symptoms of schizophrenia. However, many patients with the serious mental illness schizophrenia do not respond fully (i.e. symptoms such as delu-sions and hallucinations still remain) with a standard dose of an initially prescribed antipsy-chotic drug. In such cases, clinicians can consider increasing the antipsychotic dose beyond regular thresholds or switching to a different antipsychotic drug in order to enhance anti-psychotic efficacy. Reagarding switching strategies, no clear difference between increasing the dose of the antipsychotic drug and switching to a different antipsychotic was shown. The available evidence was extremely limited and of very low quality. Guidelines originally recommended waiting for four to eight weeks before switching to another drug, arguing that the full efficacy of a given drug is reached after a longer period of treatment [49]. Recent data suggest, however, that non‐responders can be detected as early as two weeks after initi-ation of treatment [50]. It is estimated that between one‐fifth and one‐third of patients with schizophrenia do not respond adequately to standard antipsychotic treatment[51]. There are few studies regarding switching another antipsychotics. Aripiprazole is suggested to be highly useful especially in patients in whom adverse reactions to other drugs made the con-tinuation of drug therapy difficult, but there is no anticholinergic action; therefore, anxiety or impatience suggestive of choline rebound may appear on switching from other drugs. Similarly, a method to switch other drugs to asenapine, which does not have any anticholin-ergic action, should also be reviewed.

Further, clinicians might find it rational to combine with an antidepressant for negative symptoms, benzodiazepines for comorbid anxiety or distress, or perhaps a mood stabilizer in patients suffering from incapacitating mood instability[52]. In addition, some clinicians find it rational to combine several antipsychotic drugs when aiming for a superior effect and hoping to reduce side effects. However, most of these clinician-argued reasons for antipsy-chotic combination treatment lack a clear rationale and documentation for therapeutic bene-fit [52].No concrete criteria for APP have been established, but APP was recently de-fined as the use of ≥2 antipsychotics primarily for schizophrenia treatment [8]. Con-sidering that the efficacy of APM is insufficient in patients with schizophrenia, clini-cians should primarily attempt to relieve positive. and/or negative symptoms, especially positive symptoms, by utilizing APP. Further-more, APP is also used for the treatment of specific concomitant symptoms, such as anxiety, cognitive dysfunction, impulsive/aggressive behavior, and sleep disturbance. In addition, the reasons why APP is selected include the prevention of recur-rence/recrudescence, avoidance of high-dose usage by APM, duplication on switching monotherapy, shortening of the admission period, inhibition of readmission, promo-tion of the treatment response, and prevention of adverse events based on different profiles of affinity for various receptors [53]. In addition, the appearance of antipsy-chotics with various pharmacological profiles (Table 2) has increased the number of drug options, and the number of APP variations tends to increase. Clozapine is the only drug that may be indicated for refractory schizophrenia, but it may induce a serious adverse reaction, agranulocytosis; therefore, the rate at which this drug is used is extremely low [10,54,55]. In the future, the widespread use of clozapine as a general drug therapy should be promoted [56,57]”. (Page 5 line4-Page6 line13)

Comment 5:  Proper use of antipsychotics: Ad one more table at the beginning with dose ranges and equivalent doses of all mentioned antipsychotics. In all tables should be added frequently used typical antipsychotics. This subsection is better to be divided into different paragraphs. Firstly, the individual profile of action of antipsychotics with their clinical properties needs to be more thoroughly commented on in the text, not only in the table legend. Secondly, present the possible adverse effects. Thirdly, about switching strategies. 

Response: I added the Table (Table 1. Dosage and chlorpromazine equivalents doses (CPZeq) of                                           the selected second-generation antipsychotics). (Page 4)

 And also added two antipsychotics (paliperidone and brexpiprazole) frequently used as a combination therapy in Table 1, 2 and 3.(Page 4)

 I also added 1) the individual profile of action of antipsychotics with their clinical properties in the text , 2) possible adverse effects and 3) about switching strategies as follows.

  • the individual profile of action of antipsychotics with their clinical properties in the text;

     Namely, the standardised mean differences (SMD) with 95% credible intervals, compared with placebo, in overall change in symptoms were: clozapine -0.89, -1.08 to -0.71; olanzap-ine -0.56, -0.62 to -0.50; risperidone -0.55, -0.62 to -0.48; paliperidone -0.49, -0.59 to -0.38; haloperidol -0.47, -0.53 to -0.41; quetiapine -0.42, -0.50 to -0.33; aripiprazole -0.41, -0.50 to -0.31; asenapine -0.39, -0.52 to -0.26; and brexpiprazole -0.24, -0.53 to -0.05. The SMDs that significantly reduced positive symptoms compared with placebo ranged between –0.61 (95% CrI –0.68 to –0.54) for risperidone to –0·17 (–0·31 to –0·04) for brexpiprazole [34]. Olanzapine, paliperidone, and haloperidol were significantly more effective than many other drugs [34]. The SMDs that significantly reduced negative symptoms compared with placebo ranged between –0·62 (95% CrI –0·84 to –0·39) for clozapine to –0·25 (–0·36 to –0·14) for brexpiprazole. Clozapine, olanzapine, and, to a lesser extent, risperidone reduced negative symptoms significantly more than many other drugs [34]. (Page 2 line26-Page6 line37)

  • possible adverse effects;

     The characteristics of each drug presented in Table 3 are based on relative comparison ra-ther than absolute properties of the drug. Therefore, it should be mentioned that any drug has at least all adverse reactions in the individual patient. Sedation can be a therapeutic tar-get in the acute treatment of patients presenting with agitation or severe behavioral symp-toms. Sedation is linked with the blockade of histaminergic receptors and is highest for clozapine, gradually decreasing from quetiapine, olanzapine, haloperidol, asenapine, and risperidone. In general, antipsychotic agents are associated with weight gain, and are caution for patients with diabetes mellitus. Although most high-potency FGAs has an issue ave rela-tively little weight gain potential, low-potency FGAs and most SGAs can substantially in-crease the risk of weight gain and ultimately of obesity in schizophrenia [37]. However, the weight gain potential differs substantially among SGAs, and certain FGAs can even induce more weight gain than specific SGAs[37]. In particular, clozapine and olanzapine strongly increase body weight and increase the risk of diabetes mellitus. Extrapyramidal symptoms (EPS), including bradykinesia, muscle rigidity, tremor, dystonia, akathisia, and tardive dys-kinesia, are linked to the ratio of D2 receptor to 5HT2A receptor binding [38]. The highest in-cidence of EPS in patients with schizophrenia occurs with haloperidol, with moderate EPS being observed with risperidone, paliperidone, milder EPS being observed with asenapine, ari-piprazole, and brexpiprazole. Akathisia, defined as a compelling need for constant motion, as-sociated with marching up and down, crossing and uncrossing the legs when sitting [39]. Akathisia can occur with both FGAs and SGAs, but FGAs are more likely to produce clinically relevant akathisia as per one meta-analysis [40]. Comparative meta-analyses show that aripiprazole induces more akathisia than olanzapine in schizophrenia and clozapine and risperidone oc-curs with moderate [38, 41]. It should be noted that some overlap between akathisia and pseudo-akathisia labeling may have influenced these results. Dry mouth and constipation are anticholinergic side effects, most frequent with clozapine, olanzapine, quetiapine, and low-potency FGAs, that increases the risk of dental caries according to a large popula-tion-based study [42]. Sialorrhea is a frequent and paradoxical side effect of clozapine[43]. Hypersalivation substantially impairs quality of life and may interfere with social function-ing. The degree of hyperprolactinemia depends on the D2 receptor occupancy and on the antagonist properties of the antipsychotics [44]. Thus, antipsychotics with a higher D2 affin-ity and antagonist properties increase prolactin serum levels the most, namely, haloperidol risperidone and paliperidone [44]. Mild hyperprolactinemia is described with olanzapine and asenapine; and no hyperprolactinemia is observed with the use of quetiapin and clozapine. On the contrary, partial D2 agonists, such as aripiprazole and paliperidone, can lower prolac-tin levels, even below drug-free baseline, and adjunctive aripiprazole decrease hyperprolac-tinemia associated with other antipsychotics [45]. (Page 2 line39-Page3 line15)

  • about switching strategies;

     Reagarding switching strategies, no clear difference between increasing the dose of the anti-psychotic drug and switching to a different antipsychotic was shown. The available evi-dence was extremely limited and of very low quality. Guidelines originally recommended waiting for four to eight weeks before switching to another drug, arguing that the full effi-cacy of a given drug is reached after a longer period of treatment [49]. Recent data suggest, however, that non‐responders can be detected as early as two weeks after initiation of treatment [50]. It is estimated that between one‐fifth and one‐third of patients with schizo-phrenia do not respond adequately to standard antipsychotic treatment[51]. There are few studies regarding switching another antipsychotics. Aripiprazole is suggested to be highly useful especially in patients in whom adverse reactions to other drugs made the continuation of drug therapy difficult, but there is no anticholinergic action; therefore, anxiety or impa-tience suggestive of choline rebound may appear on switching from other drugs. Similarly, a method to switch other drugs to asenapine, which does not have any anticholinergic action, should also be reviewed. (Page 5 line10 - line22)

Comment 6: Polypharmacy: Explain treatment resistance (nonresponse). Other options for augmentation with psychotropic drugs before APP. When is APP justified and in which combinations (FGA/SGA)? 

Response: I added the Figure 1 (clinical steps in APP) and 1) Explain treatment resistance (nonresponse), 2) Other options for augmentation with psychotropic drugs before APP, 3) When is APP justified and in which combinations (FGA/SGA)? as follows.

  • Explain treatment resistance (nonresponse);

Recent data suggest, however, that non‐responders can be detected as early as two weeks after initiation of treatment [50]. It is estimated that between one‐fifth and one‐third of patients with schizophrenia do not respond adequately to standard antipsychotic treatment [51]. (Page 5 line 15 - line 17)

  • Other options for augmentation with psychotropic drugs before APP;

Further, clinicians might find it rational to combine with an antidepressant for negative symptoms, benzodiazepines for comorbid anxiety or distress, or perhaps a mood stabilizer in patients suffering from incapacitating mood instability[52]. (Page 5 line 23- line 25)

3) When is APP justified and in which combinations (FGA/SGA)?

Figure 1 (clinical steps in APP) and “In addition, some clinicians find it rational to combine several antipsychotic drugs when aiming for a superior effect and hoping to reduce side effects. However, most of these cli-nician-argued reasons for antipsychotic combination treatment lack a clear rationale and documentation for therapeutic benefit [52]. No concrete criteria for APP have been estab-lished, but APP was recently defined as the use of ≥2 antipsychotics primarily for schizophrenia treatment [8]. Considering that the efficacy of APM is insufficient in pa-tients with schizophrenia, clinicians should primarily attempt to relieve positive and/or negative symptoms, especially positive symptoms, by utilizing APP. Further-more, APP is also used for the treatment of specific concomitant symptoms, such as anxiety, cognitive dysfunction, impulsive/aggressive behavior, and sleep disturbance. In addition, the reasons why APP is selected include the prevention of recur-rence/recrudescence, avoidance of high-dose usage by APM, duplication on switching monotherapy, shortening of the admission period, inhibition of readmission, promo-tion of the treatment response, and prevention of adverse events based on different profiles of affinity for various receptors [53]. In addition, the appearance of antipsy-chotics with various pharmacological profiles (Table 2) has increased the number of drug options, and the number of APP variations tends to increase. Clozapine is the only drug that may be indicated for refractory schizophrenia, but it may induce a serious adverse reaction, agranulocytosis; therefore, the rate at which this drug is used is extremely low [10,54,55]. In the future, the widespread use of clozapine as a general drug therapy should be promoted [56,57].” (Page 5 line 25 – Page6 line 13)

Comment 7: Switching to APM: Consider excluding or transferring into the Discussion the following part: “Concerning APM, generally, the costs… than that to APP (63).”

Response: I transferred the following part: “Concerning APM, generally, the costs… than that to APP (90)” into the end of paragraph. (Page 8 line 48 - Page 9 line 3)

Comment 8: Practical use of dosage forms: It is important to comment on the combination of LAI and other AP. Transfer the part about formulations in the section Proper use of antipsychotics and add to all of this about prolonged-release tablets and inhaling devices.

 Response: I added the comment “The efficacy and safety of LAI antipsychotic monotherapy appeared similar to the combination of LAI and other oral antipsychotics in patients with schizophrenia. Therefore, the combination of LAI and oral antipsychotics, which is commonly used in clinical practice, may not be necessary[99].” in the section of Practical use of dosage forms. (Page 9 line 44 - line 47)

And I transferred the part about formulations “When introducing long-acting injection (LAI) of antisychotics, it is necessary to administer an oral preparation of the same drug for the confirmation of tolerance (concerning paliperidone palmitate, risperidone is available). However, titer conver-sion from an oral preparation does not apply to all patients; therefore, much attention must be paid to overdosage related to switching to LAI or unexpected adverse reac-tions [46]. An inhaled antipsychotic loxapine provides a novel new option for use in the acute treatment of agitation in patients with schizophrenia, combining a rapid on-set of effect with a noninvasive route of administration [47]. Although simple to self-administer, inhaled loxapine requires a degree of cooperation from the recipient and thus will not be a substitute for an injection during psychiatric emergencies when the patient is actively refusing medication treatment [47]. The efficacy and safety of inhaled loxapine in elderly patients and in outpatient care settings remain to be estab-lished. Blonanserin transdermal patch, which is also offer potential benefits, including improved adherence, improved the symptoms of acute schizophrenia with acceptable tolerability, the most common adverse effects reported were application-site erythema and pruritu [48].” (Page 3 line 17 - line 31).

 I was not able to find the study regarding prolonged-release tablets with schizophrenia.

Comment 9: Conclusion: Should be rewritten following all changes made. It should reflect the significant points of the manuscript. 

Response:  I revised Conclusion section as follows; “APM is recommended as the gold standard of drug therapy for schizophrenia re-gardless of its stage in clinical practice. However, there are many cases in which there is no response to APM at a sufficient dose or only a partial response is achieved. The current studies provided the merits and limitations of APP, which is recommended as an option for non-responders to APM, as well as strategies to overcome the limitations. In any case, it is necessary to understand the pharmacological characteristics of vari-ous antipsychotics/expected adverse reactions and carefully review the indication of APP, considering drug information based on experience regarding their use in indi-vidual patients. In particular, when indicating APP for elderly patients, attention must be paid to various adverse events related to antipsychotics, considering concomitant physical diseases, aging-related physiological hypofunction, and combined drug inter-actions. In addition, even if APP is reached, it should not be chronically continued, but physicians must make efforts to promote APM while monitoring the balance between the effect and adverse event precisely. Strategies to avoid APP include dose-/drug-reduction of antipsychotics using the SCAP method and the practical use of dosage forms, such as LAI. The final goal of schizophrenia treatment is to achieve rehabilitation [100]. The continuation of drug therapy is the most important strategy for preventing recurrence, readmission, suicide attempts, or impulsive behaviors and maintaining an improvement. In the future, further evidence to indicate APP for pa-tients with schizophrenia, including elderly patients, may be necessary." (Page 10 line 2- line 19)

Comment 10: There are a lot of newer references in the field.

Response:  I added several references.

Comment 11: Add a figure reflecting clinical steps in APP. 

Response:  I added “Figure 1. clinical steps in APP”. (Page 5)

Round 2

Reviewer 2 Report

Reference for extended-release AP are existing, for example: Wang D et al. J Clin Psychopharmacol. 2022 Jul-Aug 01;42(4):383-390. doi: 10.1097/JCP.0000000000001573. 

Language should be additionally improved.

Author Response

Responses to Reviewer 2 comments:

I wish to express our appreciation to the Reviewer for his or her insightful comments, which have helped me to significantly improve the quality of our manuscript.

Comment: Reference for extended-release AP are existing, for example: Wang D et al. J Clin Psychopharmacol. 2022 Jul-Aug 01;42(4):383-390. doi: 10.1097/JCP.0000000000001573. 

Language should be additionally improved.

Response: In accordance with the Reviewer’s comment, I added the references including related references and added the following sentences. “Extended-release paliperidone is a new atypical antipsychotic chemically related to the well-known antipsychotic risperidone. It has been formulated in an osmotic controlled-release oral delivery system that minimizes peak-trough fluctuations and, by obviating dose-titration, allows once-daily dosing with a therapeutically active dose from the first day [49]. However, it was observed that long-term treatment of extended-release paliperidone as well as olanzapine caused significant increase in weight and waist circumference [50, 51]. These studies reinforce the necessity of regularly monitoring metabolic parameters in patients with schizophrenia taking atypical antipsychotics, including extended-release paliperidone. (Page 3 line 31-line 38)

Further, the language of the paper will be corrected by a native speaker, and the language corrected version will be sent next week.
